# Biosiliceous and geochemical response to biotic and climatic events in the Palaeocene

Cécile Figus[1,2], Steve Bohaty[3], Johan Renaudie[4], and Jakub Witkowski[1]

[1]Institute of Marine and Environmental Sciences, University of Szczecin, 70-383 Szczecin, Poland
[2]Doctoral School, University of Szczecin, 70-383 Szczecin, Poland
[3]Institute of Earth Sciences, University of Heidelberg, 69120, Heidelberg, Germany
[4]FB1 Dynamik der Natur, Museum für Naturkunde, 10115 Berlin, Germany

**Correspondence:** Cécile Figus (cecile.figus@phd.usz.edu.pl)

**Abstract.** Hyperthermal events are a key element in understanding Palaeogene climate history, but many of these events other than the prominent Palaeocene Eocene Thermal Maximum are poorly understood and studied. Two hyperthermal events that occurred in the middle to late Palaeocene include the Latest Danian Event (LDE) and the Early Late Palaeocene Event (ELPE). Most studies of these events focus on calcareous nannofossils and foraminifera, as well as geochemical data and astronomical

tuning, but, to date, none consider biosiliceous production and flux. We therefore present eight records of biosiliceous fluxes, supported by geochemical data, from the Atlantic, Pacific and Indian ocean sites spanning parts of the Palaeocene. Our results show pronounced variability in biosiliceous fluxes through the Palaeocene, with a peak at the time of the LDE. Establishing a link between the ELPE and biosiliceous flux variability through this time interval is more challenging, but the occurrence of peaks in biosiliceous fluxes after this event may indicate a global response of biogenic silica to several short-term biotic events,

including the ELPE.

## 1   Introduction

The study of Palaeogene climates (~66 to ~23 Ma) includes a focus on intense short-term (up to ~400,000 years; Barnet et al., 2019) climatic events called 'hyperthermals', which are often compared to the present-day climate change in terms of greenhouse gas emissions into the atmosphere (Foster et al., 2018; Kiessling et al., 2024). These hyperthermal events are

associated with, among other characteristics, rapid (between 1,000 and 100,000 year duration) increases in temperature and atmospheric $CO_2$, as well as ocean acidification (Bowen et al., 2004; Foster et al., 2017; Westerhold et al., 2018). The most widely known hyperthermal event is the Palaeocene-Eocene Thermal Maximum (PETM; ~56 Ma) (Kennett and Stott, 1991; McInerney and Wing, 2011), when large volumes of isotopically depleted carbon were released into the oceans and atmosphere over the course of ~10,000 years (Penman et al., 2014). Although the PETM is a hallmark of Palaeogene greenhouse climates,

it is not the only short-lived climate event to have occurred during the Palaeogene, which was initially characterized by extreme warmth following the Cretaceouus/Palaeogene (K/Pg) massive extinction event, before transitioning to global cooling from ~49 Ma onwards. Several climatic events occurred during the early Palaeogene (Littler et al., 2014; Barnet et al., 2019; Westerhold et al., 2020) such as the Early Eocene Climate Optimum (EECO; ~53 to ~49 Ma) (Zachos et al., 2008).

Two short-lived climatic events, which are particularly understudied, occurred in the Palaeocene (~66 to ~56 Ma): the Latest
Danian Event (LDE; ~62.2 Ma) and the Early Late Palaeocene Event (ELPE, also known as the Mid-Palaeocene Biotic Event
or MPBE; Bernaola et al., 2007). Published ages for the ELPE range from ~58.4 Ma (Bralower et al., 2002; Petrizzo, 2006) to
~59.5 Ma (Li et al., 2024). In addition, the qualification of the ELPE as a hyperthermal event has long been debated (Littler
et al., 2014), as it is characterized by multiple changes in the biota and environment (Coccioni et al., 2019), but evidence for a
negative carbon isotope excursion (CIE) or warming is not present in all $\delta^{13}C$ and $\delta^{18}O$ records spanning this interval (Hollis
et al., 2022; Li et al., 2024). The LDE, in contrast, is associated with the largest negative CIE in the Palaeocene (Bornemann
et al., 2009; Dinarès-Turell et al., 2012), and has been linked to enhanced volcanic activity (Jehle et al., 2015).

Reconstructing long- and short-term palaeoclimatic changes is possible using numerous proxies (e.g., Froelich and Misra,
2014; Jehle et al., 2015; Westerhold et al., 2018, 2020), such as $\delta^{18}O$ (for ice volume and temperature) or $\delta^{13}C$ (for the
carbon cycle perturbations), but also through the use of biogenic opal records. Fossil diatoms and radiolarians are remnants of
skeletons composed of opal, amorphous or crystallized depending on the stage of diagenetic transformation (Rice et al., 1995;
Yanchilina et al., 2020), and thus are part of the biosiliceous fraction contained in marine sediments. Based on a low-resolution
compilation of data from deep-sea diatom-bearing sites in the Atlantic, Pacific and Indian oceans, Figus et al. (2024b) indicate
no apparent biosiliceous flux response to Palaeocene hyperthermal events, while the stratigraphic distribution of shallow marine
diatomite deposited in epicontinental seas during this period does suggest a link between certain hyperthermal events (e.g., the
PETM) and diatomite accumulation (Figus et al., 2024a). However, compiling stratigraphic data from deep-sea sites drilled by
the Deep Sea Drilling Project (DSDP) campaigns and its successors has revealed an uneven distribution of the number of sites
containing Palaeogene biosiliceous sediments, with a lower number of sites covering the Palaeocene, as well as a bias in the
preservation of early Cenozoic siliceous microfossils (Figus et al., 2024b). We therefore decided to focus on Palaeocene signals
(between ~63 and ~62 and between ~60 and ~57 Ma) in opal fluxes and geochemical data at eight deep-sea sites drilled in the
Atlantic, Pacific and Indian Ocean, that do preserve Palaeocene-aged diatom and radiolarian assemblages. The new records
generated for this study are compared with published data to reconstruct the trends in biosiliceous fluxes across these two
intervals, and to investigate a possible link with the LDE and ELPE.

## 2 Materials and methods

### 2.1 Site selection

The eight sites selected for this study are located in the North and South Atlantic, South Pacific and Indian Ocean (Table 1,
Fig. 1). We focused on sites that contain abundant and/or well-preserved diatoms as documented in DSDP and Ocean Drilling
Program (ODP) reports, including DSDP Sites 208 and 384 and ODP Sites 1050, 1051 and 1121. The Palaeocene sequences
recovered at these sites also contain abundant radiolarians over the entire section analyzed in this study.

Site selection was also determined by the stratigraphic span of the sediments, i.e., we specifically targeted intervals expected
to include a record of the LDE and ELPE. The age-depth models used for each site were updated to GTS2012 (Gradstein
et al., 2012), based on Renaudie et al. (2018) for 700B, 752A and 1121B, and Witkowski et al. (2020) for the remaining sites.

**Table 1.** Sites and Holes included in this study, with geographical coordinates, bathymetry and palaeobathymtery at 62.2Ma, location, drilling report reference and number of samples analyzed.

| Site/Hole | Coordinates | Bathymetry (mbsl) | Palaeobathymetry at 62.2Ma (mbsl) | Location | Reference | # Samples analyzed |
|---|---|---|---|---|---|---|
| 208 | 26°06.61'S, 161°13.27'E | 1545 | 1029 | South Pacific | Shipboard Scientific Party (1973) | 28 |
| 384 | 40°21.65'N, 51°39.80'W | 3909 | 3643 | North Atlantic | Shipboard Scientific Party (1979) | 87 |
| 700B | 51°31.977'S, 30°16.688'W | 3601 | 2528 | South Atlantic | Shipboard Scientific Party (1988) | 93 |
| 752A | 30°53.475'S, 93°34.652'E | 1086.3 | | Indian Ocean | Shipboard Scientific Party (1989) | 43 |
| 1050A | 30°05.9977'N, 76°14.1011'W | 2299.8 | | North Atlantic | Shipboard Scientific Party (1998a) | 106 |
| 1050C | 30°05.9953'N, 76°14.0997'W | 2296.5 | 2301 | North Atlantic | Shipboard Scientific Party (1998a) | 7 |
| 1051A | 30°03.1740'N, 76°21.4580'W | 1982.7 | 2125 | North Atlantic | Shipboard Scientific Party (1998b) | 203 |
| 1121B | 50°53.8740'S, 176°59.8620'E | 4492 | | South Pacific | Shipboard Scientific Party (2000) | 52 |

Palaeobathymetry at 62.2 Ma (LDE) was computed for Sites/Holes 208, 384, 700B, 1050C and 1051A (see Table 1) using PyBacktrack (Müller et al., 2018), along with the age models detailed above, and the detailed lithological description of each Site/Hole in their respective Initial Reports (Shipboard Scientific Party, 1973, 1979, 1988, 1998a, b; the files produced as input for PyBacktrack can be found in the Supplementary Materials).

## 2.2 Stable isotope analysis

Stable isotope analysis of bulk carbonates was carried out on samples from Holes 700B, 752A and 1121B. Each sample of bulk sediment was freeze-dried and then ground using a mortar and pestle. 50 to 100 $\mu g$ of pulverized sediment was analyzed for carbon and oxygen stable isotope ratios using a Thermo Scientific Kiel IV Carbonate device coupled with a MAT253 isotope ratio mass spectrometer at Heidelberg University (Laboratory for Stable Isotope Mass Spectrometry). The $\delta^{13}$C and $\delta^{18}$O results (expressed in per mil, ‰) were normalized and calibrated against the Vienna Pee Dee Belemnite (VPDB) standard (see Fig. 2 and full results in Supplementary Material Table S1).

## 2.3 Biogenic opal determination

The samples underwent silica extraction following the Olivarez Lyle and Lyle (2002) protocol, modified for this study (we used 1M KOH and 10 mg sediment subsamples, instead of 2M KOH and 20 mg subsamples indicated in the original protocol). This modification aims to avoid silica polymerization, as described in Witkowski et al. (2021). The concentrations of biogenic opal in each sample were then determined by heteropoly blue method (Hach method 8186) on a Hach DR-3900 spectrophotometer.

The results obtained were used to calculate the percentage weight of opal per gram of sediment in each sample (see Supplementary Material Fig. S1) and the mass accumulation rate (Fig. 2), using the methods described in Witkowski et al. (2021).

The silica extraction protocol was repeated a second time on 10 samples (see Supplementary Material Table S2), in order to verify the biosiliceous content of each sample before the addition of KOH, in the middle of the silica extraction and after the silica extraction. The contents of each samples were collected during these three phases and observed by light microscopy using a Leica DMLB microscope with a ×20 objective. The results are available in the Supplementary Material Table S2.

### 2.4   Statistical treatment

The $_{bio}SiO_2$ flux data produced for this study were linearly interpolated and smoothed using a cubic smoothing spline (Green and Silverman, 1994) in order to compare records between sites. A Pearson correlation coefficient and cross-correlation function were calculated on the smoothed $_{bio}SiO_2$ flux data for each pair of sites. In order to compare only the periods of interest for this study and to improve the significance of the results, cross-correlations were also applied focusing on the data between 60 and 57 Ma (Supplementary Material Table S3), and between 63 and 62 Ma (Supplementary Material Table S4). The

results indicate the correlation between peak $_{bio}SiO_2$ fluxes at different sites during these periods. Pearson coefficient and cross-correlation function were also calculated between $_{bio}SiO_2$ flux and percentage of $CaCO_3$ at each site (Supplementary Material Table S5), to determine whether $_{bio}SiO_2$ peaks between 60 and 57 Ma were correlated with carbonate dissolution. The comparison between time series was carried out using R (R Core Team, 2024).

### 3   Results

### 3.1   Carbon and oxygen isotopes


Stable isotopes of carbon and oxygen were measured in Holes 700B, 752A and 1121B. Each of these records spans the Selandian (~61.6 to ~59.2 Ma) and Thanetian (~59.2 to ~56 Ma), but the end of the Danian Stage (~66 to ~61.6 Ma) appears to be recorded only in Hole 700B among the samples studied here. In order to enhance the stable isotope data generatedto detect the LDE, we decided to use the $\delta^{18}O$ and $\delta^{13}C$ results of Hollis et al. (2014) for Hole 1121B, which include data for the late

Danian. Our results and those of Hollis et al. (2014) for Hole 1121B were cross-correlated (Pearson coefficient on the overall record is 0.714) to ensure compatibility between our data and this record taken from the literature.

    A cross-correlation on the portion of data from 63 to 62 Ma was performed between Holes 700B and 1121B (using data from Hollis et al., 2014), confirming the apparent alignment of the $\delta^{18}O$ and $\delta^{13}C$ peaks at (~62.2 Ma in Fig. 2c-d.

    In the period between 60 and 57 Ma, which is thought to include the ELPE, it is more difficult to distinguish a trend across

all records (Fig. 2). While the $\delta^{13}C$ results for Holes 700B and 1121B (our results and Hollis et al., 2014) show negative excursions between 59.2 and 59 Ma, Hole 752A has a peak in the opposite direction (Fig. 2a). The same difference appears in the $\delta^{18}O$ data with positive peaks for Holes 700B and 1121B, but no apparent signal for Hole 752A (Fig. 2).

**Table 2.** Table of abundances of diatoms, radiolarians, silicoflagellates and sponge spicules at Sites/Holes 208, 384 and 700B in samples corresponding to the LDE peak in biosiliceous flux. The sample at Site *208 corresponds to the drop in $_{bio}SiO_2$ flux at 62.2 Ma. A: Abundant, C: Common, F: Few and R: Rare.

| Site/Hole | Samples | Diatoms | Radiolarians | Silicoflagellates | Sponges spicules | Comments |
|---|---|---|---|---|---|---|
| *208 | 21-208-30R-5-101_102 | A | C | F | A | Dominance of diatoms with *S. turris* complex, Hemiauloides and pennates |
| 208 | 21-208-30R-6-15_16 | A | A | R | F | Dominance of radiolarians. Diatoms are mainly *S. turris* complex, Hemi-auloides, pennates and tripolar forms |
| 384 | 43-384-11R-2-100_101 | F | A | A | F | Numerous fragments of radiolarians and silicoflagellates |
| 700B | 114-700B-32R-3-50_51 | F | C | R | R | Dominance of radiolarian fragments |

### 3.2 Biogenic silica

The biogenic opal data measured in this study encompass the inferred LDE interval at only three Sites/Holes: 208 (South Pacific), 384 (North Atlantic) and 700B (South Atlantic). Figure 2 shows a peak in $_{bio}SiO_2$ flux around 62.2 Ma at Site 384 and Hole 700B, while the peak at Site 208 occurs ~100 kyrs prior (~62.3 Ma), followed by a drop in $_{bio}SiO_2$ flux around 62.2 Ma. To determine whether the peak observed at ~62.2 is present in other records, we compared our results with data from Holes 1050C and 1051A published in Witkowski et al. (2021). In both records, there is a peak in $_{bio}SiO_2$ flux at ~62.2 Ma (Fig. 2b). Furthermore, all records are positively or negatively correlated with each other over the section between 63 and 62 Ma (see Supplementary Material Table S4), corroborating the occurrence of an event triggering an increase in $_{bio}SiO_2$ flux at ~62.2 Ma.

The microfossil content (Table 2) of samples corresponding to peaks in biosiliceous flux (Fig. 2) at Sites/Holes 208, 384 and 700B shows a dominance of radiolarians over diatoms, whereas the sample from Site 208, which corresponds to the drop in biosiliceous flux at ~62.2 Ma, is dominated by diatom assemblages. In addition, observation of the sample contents pre-, mid- and post-silica extraction revealed that all siliceous microfossils (see Supplementary Material Table S2) were already dissolved after six hours of protocol (mid-silica extraction), which means that our biosiliceous flux results are not biased by the presence of non-dissolved siliceous microfossils.

The Selandian-Thanetian boundary ( ~59.2 Ma) is also covered by our bioSiO2 results from Site 384 and Hole 700B, as well as from Holes 752A and 1121B (Fig. 2). In addition, we have used the published data (Witkowski et al., 2021) from Holes 1050A and 1051A to compare with our results (Fig. 2b). However, the trends in the results are more difficult to determine between 60 and 57 Ma than between 63 and 62 Ma. An increase in bioSiO2 flux occurs around 59 Ma at Site 384 and Holes 700B and 1121B (Fig. 2). At Holes 752A and 1051A, the biogenic opal peak happens later, between ~58.8 and ~58.5 Ma,

while at Hole 1050A, the peak occurs before ~59.2 Ma (Fig. 2a-b). Cross-correlations of the data between 60 and 57 Ma (Supplementary Material Figs Table S3) show that there are correlations between each site's results, but the value of the delay
between each peak is significant (i.e., low Pearson coefficients and high lag values).

## 4 Discussion

The LDE and ELPE are relatively understudied, with less literature available than for the PETM, for example. However, both events have been consistently recorded in deep-sea sediments, enabling a good temporal delimitation (Littler et al., 2014; Jehle et al., 2015). The negative carbon excursion that defines the LDE occurred at the end of the Danian, around 62.2 Ma
(Bornemann et al., 2009; Dinarès-Turell et al., 2012). The ELPE, on the other hand, corresponds to a carbonate dissolution event at the Selandian–Thanetian boundary, which is astronomically calibrated (Westerhold et al., 2008; Hilgen et al., 2015).

Most studies of these events focus on calcareous nannofossils and foraminifers (e.g., Petrizzo, 2006; Sprong et al., 2012; Jehle et al., 2015; Alegret et al., 2016), or on geochemical data and astronomical tuning (e.g., Westerhold et al., 2008; Littler et al., 2014; Hilgen et al., 2015; Li et al., 2024), but, to date, siliceous microfossils and biosiliceous fluxes have not been
investigated with regard to these two events. Analysis of the biogenic opal contained in Palaeocene sediments may provide new information on these events and their impact on the silicon cycle.

### 4.1 Biosiliceous and geochemical response to the LDE

In addition to analyzing biosiliceous fluxes, it is important to verify the presence of a stable isotope response to detect the LDE in our records. The isotopic results from Holes 700B and 1121B for the LDE interval are consistent with the negative CIE (Fig.
2c-d) and a short warming episode described in previous studies (Jehle et al., 2015, 2019; Alegret et al., 2016). Furthermore, although there is no isotopic record for Site 384 in our study, it is interesting to note that Haq et al. (1979) explain that while calcareous nannofossil assemblages indicate a cooling trend between ~61 and ~58 Ma at Site 384, assemblages found around ~62 Ma suggest warming, which likely corresponds to the LDE.

All the $_{bio}SiO_2$ flux records (Fig. 2) show a peak around the LDE (~62.2Ma), except for Site 208 (Fig. 2d) which displays
a peak before the LDE (~62.3Ma), followed by a drop at the time of the event. One possible explanation could be a problem in the calibration of the data to GTS2012, causing a mismatch in the position of the peak. This hypothesis is supported by the cross-correlation results, which show that the peaks in $_{bio}SiO_2$ fluxes at each site throughout the LDE are correlated positively or negatively, as explained in the Results section, and the fact that the age-depth model for Site 208 is very coarse. However, we also consider the plausibility of a palaeoceanographic reason for this temporal offset between Site 208 and the
other Sites/Holes. The intense carbon release and enhancement of the hydrological cycle associated with these climatic events, assuming a linear model for the silicate weathering feedback strength, should be expected to enhance the chemical alteration and mechanical erosion of rocks on land, and thus to manifest as elevated rates of silicate weathering (Berner et al., 1983; Penman et al., 2019). The resulting input of dissolved silica—used by siliceous biota to build their exoskeletons—into the oceans is increased (Penman, 2016), leading to a rise in the number of siliceous microfossils in sediments. Palaeobathymetries

calculated with PyBacktrack (Müller et al., 2018) for the LDE show that Site 208 was shallower and closer to the coast than the other Sites/Holes at that time (see Table 1). Site 208 would therefore have received input from silicate weathering more quickly than the Sites/Holes further from the shore. In addition, the time spent by dissolved silica in the water column before being buried would be longer for the deeper Sites/Holes than for Site 208. Nevertheless, it is also possible that the peak in biosiliceous flux (Fig. 2d) is not a response to the LDE but to an unidentified local event.

One final question remains regarding our results: why are these peaks not detected in the global compilation of deep-sea diatom-bearing sediments of Figus et al. (2024b)? The most plausible answers would be 1) the resolution of the dataset in Figus et al. (2024b) and 2) the composition of the biosiliceous content analyzed in the present study. In Figus et al. (2024b), the data are computed with a temporal resolution of one million years, whereas the results of the current study are more precise. In addition, Figus et al. (2024b) only consider the number of sites containing diatom-bearing sediments. The $_{bio}SiO_2$ fluxes measured in this study are indifferent to the siliceous content of the sediments, whether they contain diatoms, radiolarians, or any other siliceous microfossil. It is therefore likely that radiolarians, more than diatoms, responded to the LDE. This hypothesis is supported by the microfossil content (Table 2) of the samples corresponding to the peaks in $_{bio}SiO_2$ fluxes at ~62.2 and ~62.3 Ma, which indicate a dominance of radiolarians over diatoms during the LDE.

## 4.2 Characterization of the ELPE

### 4.2.1 A climatic and/or biotic event?

The ELPE was described early on in the scientific drilling literature without being named, for example in the DSDP report for Site 384, which describes a change in palaeoproductivity and a carbonate dissolution event (Shipboard Scientific Party, 1979). It has since been better identified with precise characteristics. To improve the identification of the ELPE, Petrizzo (2006) indicated a sequence of stages based on the study of calcareous nano-/microfossil assemblages at Shatsky Rise (northwest Pacific). These stages include the first occurrence and increase in abundance of the nannofossil *Heliolithus kleinpellii*, variations in the abundance of the planktonic foraminifera *Igorina tadjikistanensis* and *Igorina albeari*, a peak in magnetic susceptibility, and the deposition of phillipsite, a member of the zeolite group. In addition, the event seems to be associated with carbonate dissolution, probably related to a shoaling of the carbonate compensation depth (CCD) and the lysocline (Bralower et al., 2002; Petrizzo, 2006). Littler et al. (2014) explain this shoaling by a massive input of isotopically depleted carbon into the oceans during the ELPE. Bernaola et al. (2007) make the same observation, based on the occurrence of a negative CIE and a decrease in the $\delta^{18}O$ record at Zumaia, an onshore section in the Pyrenean basin. Several other studies report a negative ELPE-related CIE in shallow marine or terrestrial environments such as the eastern Tethys (Sarkar et al., 2022) and western Neo-Tethys (Coccioni et al., 2019), or two CIEs, as recorded in the Tethys Himalaya (Li et al., 2024) and northwest Argentina (Hyland et al., 2015). In the deep-sea, CIEs are reported from ODP Legs 198 (Shatsky Rise) and 208 (Walvis Ridge, South Atlantic) (Littler et al., 2014; Hilgen et al., 2015). Whereas these papers are in general agreement with the interpretation of the ELPE as a hyperthermal event, Hollis et al. (2014) provide a different explanation, correlating the ELPE with the onset of the Palaeocene Carbon Isotope Maximum (PCIM), and a climate cooling. According to Hollis et al. (2014), the decrease in carbonate content

at ODP Hole 1121B could be the result of regional cooling, linked to glacio-eustatic factors in the Antarctic region, enhancing upwelling and marine productivity during the ELPE.

### 4.2.2 Dating the ELPE

Although the ELPE is well constrained in the deep-sea, onshore sites appear to disagree with the timing of the event. At deep-sea sites, biostratigraphic and geochemical data first suggested that the ELPE occurred at ~58.4 Ma in the northwest Pacific (Bralower et al., 2002; Petrizzo, 2006) or ~58.9 Ma at South Atlantic sites (Littler et al., 2014). Revisions of these data by cyclostratigraphy, using magnetic susceptibility and iron content, place this event earlier, at the Selandian–Thanetian boundary, at ~59.2 Ma (Westerhold et al., 2008; Hilgen et al., 2015). Onshore sections also give different results, notably with the double isotopic excursion at ~59.3 and ~59 Ma in the Tethys Himalaya (Li et al., 2024) and northwest Argentina (Hyland et al., 2015). Whereas all these studies agree on the short duration of the ELPE ($< 1$ million year), the difference between ages attributed to onshore and deep-sea sites highlight questions about the correlation between shallow and deep-sea carbonate sections.

### 4.3 Biogenic silica accumulation in the early late Palaeocene

In order to assess whether a signal corresponding to the ELPE is present in our records, we investigated the original drilling reports to find evidence of the various characteristics indicative of this event. Six Sites/Holes cover the presumed ELPE stratigraphic interval: 384, 700B, 752A, 1050A, 1051A and 1121B. In the North Atlantic, only Site 384 includes well-documented (Shipboard Scientific Party, 1979) record of a potential event occurring around the Selandian–Thanetian boundary. The report indicates that around this period, smear slide data reveal the presence of 1 to 2% of amorphous iron oxide and/or haematite in the nannofossil ooze, which enhances iron concentrations in the sediments. A severe foraminiferal dissolution event is also reported in Cores 384-8R and -9R, but this does not appear to affect nannofossils, hence the lack of prominent carbonate dissolution in Fig. 3. This event occurs before the peak in biosiliceous flux (Fig. 2b), in core section 384-7-6, which precedes a positive carbon excursion at ~57 Ma, accompanied by a drop in temperature (Boersma et al., 1979). This positive CIE could represent the onset of the PCIM, with the biosiliceous flux peak (Fig. 2b) occurring between the foraminiferal dissolution interval and the PCIM. In Hole 1051A, carbonate dissolution (Fig. 3) also precedes the peak in biosiliceous content (Fig. 2b) and occurs at the same level as the peaks in magnetic susceptibility and iron, used by Westerhold et al. (2008) to determine the ELPE at this site. However, in the nearby Hole 1050A, $_{bio}SiO_2$ flux (Fig. 2b) and magnetic susceptibility (Shipboard Scientific Party, 1998a) both increase with decreasing $CaCO_3$ percentage (Fig. 3).

In the South Atlantic, the increase in $\delta^{13}C$ at Hole 700B occurs after the peak of biosiliceous flux (Fig. 2c), as at Site 384. The resolution of the % $CaCO_3$ record in Fig. 3 is too coarse to define a proper dissolution interval, but (Shipboard Scientific Party, 1988) report that the carbonate dissolution event takes place in Core 700B-28R, as does the peak in biosiliceous content (Fig. 2c), and that benthic foraminiferal assemblages disappear from Cores 29R to 26R, while siliceous microfossils are abundant in Cores 32R to 26R. Furthermore, at the depth of the biosiliceous peak, the authors report the presence of anomalies in the gamma-ray record, with an increase in radioactivity derived mainly from uranium. Gamma-ray values are also high at Hole

752A (Indian Ocean), probably due to the presence of clay minerals derived from ash alteration (Shipboard Scientific Party, 1989), except in the core where the biosiliceous peak occurs (Fig. 2a). At this site, an abrupt shift in isotopic values occurs when biosiliceous flux values are the highest (Fig. 2a) and carbonate content is reduced (Fig. 3).

For Hole 1121B, each proxy record displays several peaks (Figs 2d) just after the Selandian–Thanetian boundary. Two carbonate dissolution events appear to have occurred (Fig. 3), along with negative carbonate isotopic excursions and peaks in $_{bio}SiO_2$ content (Fig. 2d). Shipboard Scientific Party (2000) interpret the decrease in carbonate content as the result of CCD shoaling, possibly due to an inflow of cold-water masses, corrosive to carbonates.

Although some of the stages associated with the ELPE are present at the sites discussed in this study, the stratigraphy and/or identification of the event appear to be uncertain at our sites. This is highlighted by the results of the cross-correlations (see Supplementary Material Table S3), showing that peaks between 57 and 60 Ma are correlated, but not precisely aligned with each other. Furthermore, the increase in biosiliceous flux does not always occur at the time of the carbonate dissolution, but sometimes precedes it, as indicated by the Pearson coefficients (Pearson coefficient between $_{bio}SiO_2$ and $\%CaCO_3$: 0.462 at Site 384, -0.086 at Hole 700B, -0.646 at Hole 752A, -0.712 at Hole 1050A, 0.047 at Hole 1051A and -0.297 at Hole 1121B). The various results produced for this study are therefore not sufficiently significant to clearly link the increase in $_{bio}SiO_2$ to the ELPE, but it seems that siliceous microfossils became abundant after the Selandian–Thanetian boundary, for a short period of time. The differences in timing of the ELPE between sites could potentially be related to an error in the calibration of the updated age-depth models, or reveal the occurrence of several short-lived ocean changes during this interval. However, such a calibration error is a less plausible explanation, as it would have to occur in each of the eight age-depth models to produce these offsets, and the results are not mismatched in this way for the LDE.

# 5 Conclusions

The peaks in stable isotope records and biosiliceous fluxes in the Atlantic and South Pacific at the end of the Danian appear to be consistent with the occurrence of the LDE. Around the Selandian-Thanetian boundary, however, the appearance of several peaks in biosiliceous fluxes, as well as in oxygen and carbonate records, does not allow a clear correlation between the results and the ELPE. The most plausible explanation is palaeogeographic. Several short-lived biotic events may have occurred between 60 and 57 Ma, during which siliceous microfossils became dominant in the water column. Furthermore, palaeogeography may also be responsible for the temporal offset in biosiliceous flux increases between Site 208 and other Sites/Holes during the LDE. Palaeobathymetry suggests that the earlier response of biogenic silica to the LDE at Site 208 might be related to the shallower depth of this site compared to others.

Finally, this study suggests that, unlike diatoms, radiolarians appear to respond to Palaeocene climatic events, such as the LDE, and seem to be responsible for increases in bioliceous fluxes in Palaeocene deep-sea sediments. These results offer for the first time a general idea of the biosiliceous flux response to the LDE and ELPE, but a more detailed study comparing biosiliceous fluxes and siliceous microfossil diversity during Palaeocene climatic and biotic events would be necessary to

better understand how siliceous microfossils interacted with the Palaeogene climates. Furthermore, comparing shallow and deep marine environments may provide new insights into the impact of palaeoceanography on Palaeocene silicious biota.

255 *Data availability.* The results of biogenic opal determination and isotopic measurements can be found in the Supplementary Material Table S1.

*Author contributions.* JW designed the study. CF and JW prepared the samples and carried out the biogenic opal determination. SB carried out the stable isotope analysis. JR processed the data. All co-authors participated in the interpretation of the data. CF prepared the paper, with contributions from all co-authors.

260 *Competing interests.* The authors declare that they have no conflict of interest.

*Acknowledgements.* We would like to thank Danuta Cembrowska-Lech for her advice on statistical treatment, and the two anynomous reviewers who greatly contributed to the enhancement of this manuscript.

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

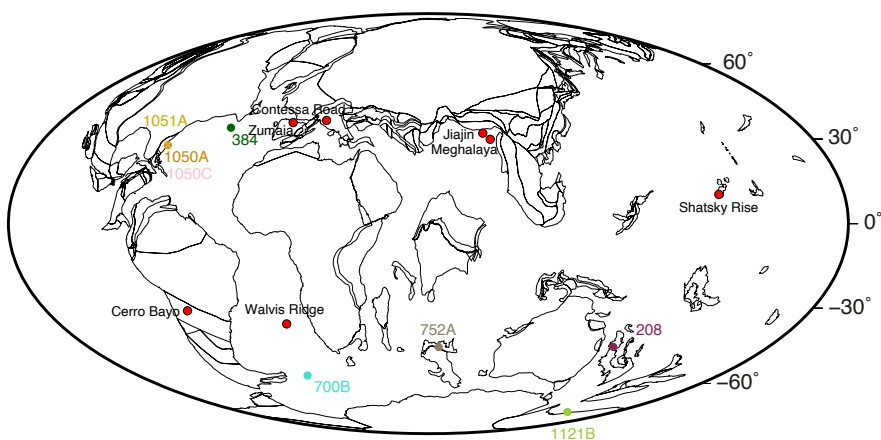

**Figure 1.** Palaeogeographic map of study Sites/Holes at ~60 Ma, with the locations of sections where the ELPE has been previously identified (Walvis Ridge in the South Atlantic, Shatsky Rise in the northwest Pacific, Zumaia in the Pyrenees, Contessa Road in the western Neo-Tethys, Jiajin in the Tethys Himalaya, Meghalaya in the eastern Tethys and Cerro Bayo in northwest Argentina). Map generated on the Ocean Drilling Stratgraphic Network Advanced Plate Tectonic Reconstruction service (www.odsn.de).

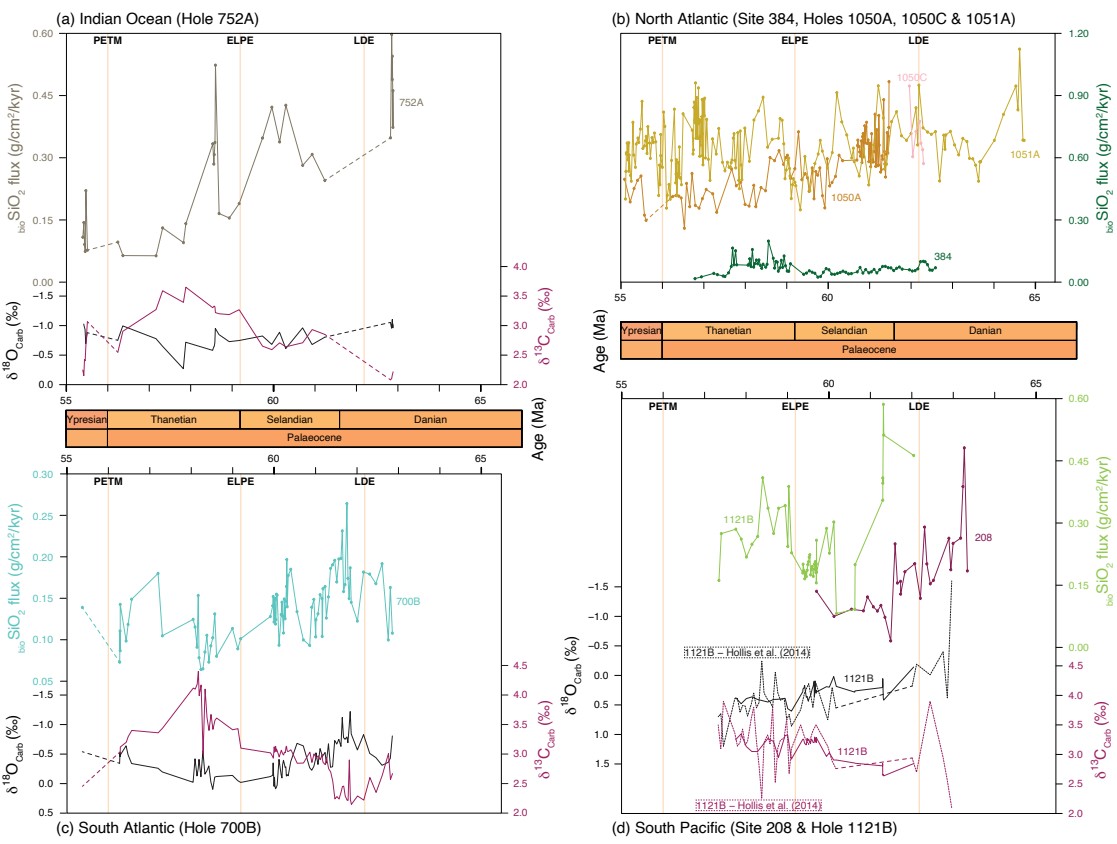

**Figure 2.** $_{bio}SiO_2$ fluxes ($g.cm^{-2}.kyr^{-1}$) and stable isotope records (in ‰) measured at study Sites/Holes in the (a) Indian Ocean, (b) North Atlantic, (c) South Atlantic and (d) South Pacific. $_{bio}SiO_2$ flux results for Holes 1050A, 1050C and 1051A are from Witkowski et al. (2021) and stable isotope records from Hollis et al. (2014) are added to the results for Hole 1121B. PETM: Palaeocene Eocene Thermal Maximum (~56 Ma), ELPE: Early Late Palaeocene Event (~59.2 Ma), LDE: Latest Danian Event (~62.2 Ma).

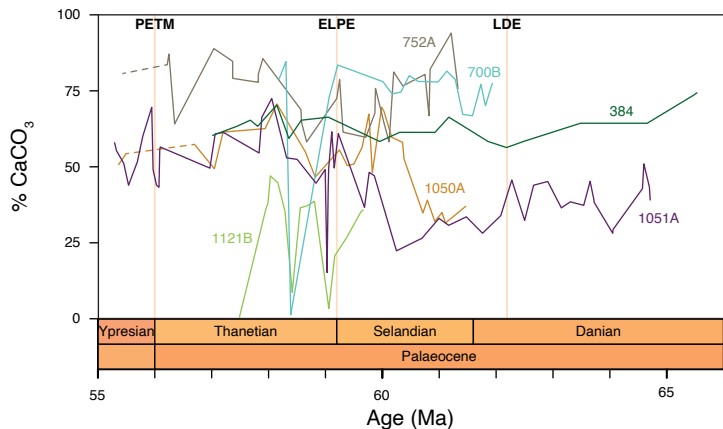

**Figure 3.** Percent $CaCO_3$ values at the study Sites/Holes from the International Ocean Discovery Program Janus database (web.iodp.tamu.edu).