# Peer review of "Biosiliceous and geochemical response to biotic and climatic events in the Palaeocene"

_EGUsphere, 2025_

## Author Response (AR1)

INSTITUTE OF MARINE
AND ENVIRONMENTAL SCIENCES
UNIVERSITY OF SZCZECIN

Climate of the Past Editorial Board
Copernicus Publications
Bahnhofsallee 1e
37081 Göttingen
Germany

Szczecin, May 19, 2025

Dear Dr. Voelker,

We thank you for the opportunity to revise our manuscript entitled 'Biosiliceous and geochemical response to biotic and climatic events in the Palaeocene'. Please find below detailed responses to the reviewer's suggestions. We have also considered the comment from Copernicus and removed the figures that made the Supplementary Material difficult to navigate, to replace them with tables of results.

Yours sincerely,
Cécile Figus, on behalf of all Co-Authors

[Figure]

**Responses to Anonymous Referee #1:**

Figus et al., presents an important study combining biosiliceous and geochemical records to help understand hyperthermal events in the Palaeocene-Eocene Thermal Maximum. Providing Carbon and Isotopes from bulk carbonates from Sites 700B, 752A, and 1121B. Additionally, they present biogenic silica weight percentage, and biogenic silica fluxes (g/cm2/kyr) from Sites 208, 384, 700B, 752A, and 1121B.

I applaud the authors for their work compiling these observations and working towards a better understanding of these hyperthermal events using isotopes and biogenic silica data. This work is highly important and draws attention to the need to better study and understand biosiliceous sedimentation during the Paleocene and the potential differences between calcareous and siliceous plankton during these events.

I offer additional comments below for the authors to consider for clarity and overall flow, as well as for incorporating your findings into this gap in biosiliceous work. I recommend adding a dedicated results section. In this section, highlight what observations were made by Site, how many samples were analyzed, and trends in the data/isotope stratigraphy (and need to incorporate other records). From this build a discussion based on how these results compare to the sites and events you wish to discuss. Additionally, the conclusion needs to be restructured and rewritten to highlight what was done, what you observed, how these relate to the events during the Paleocene, and maybe ending with a discussion of future work if you think it is merited or challenges in this kind of work moving forward.

We have added a dedicated Results section detailing the observations made for each site and the trends in the data. The number of samples has been added to Table 1 in Material and Methods. We have reworked the discussion and conclusion to improve the structure of the article and highlight what we have done.

I recommend major revisions for this work and thank the authors for their significant contributions to using biosiliceous fluxes and isotopes to better understand these hyperthermal events.

Suggested edits:

I recommend breaking up the currently mixed "Results and interpretation" section into a results section and then a discussion section. Results: Carbon and oxygen isotopes (for what sites, sample resolution, and trends). For biogenic silica values, how did you calculate the fluxes, and what are the trends in the data? Discussion: LDE focus on the geochemistry and the biogenic silica data. Next, the ELPE, and maybe start with dating the event, then characterization of the event, and finally biogenic silica in the ELPE.

We have divided the Results and Interpretation section into two sections.

Additionally, conclusions need to discuss what you did, important findings (for each record presented in this study), and potentially address challenges or things needed to better understand these events.

[Figure]

We have reworked the conclusion to better discuss what we have done, what is important in our results and some perspectives for future studies to improve our understanding of the link between Palaeocene climatic events and siliceous microfossils.

Line Comments:

Line 19—21: "characterized initially, by extreme warmth following the Cretaceous-Paleogene (K/Pg) mass extinction event, before transitioning to global cooling around 53 Ma onwards." Or something like this. Just to help with flow.

This has been modified.

Line 36: Do you mean Figus et al. (2024)? Or Figus et al. (this study)? For consistency, you should clarify what you mean

We sincerely apologise for the confusion with 'Figus et al.', it seems that LateX did not take into account the full reference during the final formatting of the manuscript. Figus et al. in the text refers to the preprint 'Controls on Palaeogene deep-sea diatom-bearing sediment deposition and comparison with shallow marine environments' by Figus et al. (2024). This has been corrected.

Line 42: Figus et al. 2024?

Same as for line 36.

Line 47: For this section you have Site selection. Could you also include number of samples or include that in your methods section for Stable isotope analysis and biogenic silica.

The number of samples has been added to the Table 1.

Line 62: I'm curious to know if these samples were checked pre- and post-dissolution/obtaining the measurements to see what remains after these dissolution experiments. I agree with using the KOH method, but sometimes, for biogenic silica dissolution, this could be biased to just diatoms, radiolarians, or sponge spicules. This doesn't need to be done on all samples, but provides a bit more information on what this opal signal is recording.

We proceeded to check 10 samples pre-, mid- and post-dissolution and added the results to the revised manuscript and Supplementary Material. The samples analysed correspond mainly to peaks in biosiliceous flux. After 6h (mid-protocol) in the heating bath with KOH, all the microfossils had already dissolved. We therefore think that the siliceous microfossils did not bias our results.

Line 79 and 80: Are there particular R packages you used for this work? Do they need to be cited?

Only the basic R functions were used.

Line 81 to 86: This paragraph needs restructuring to improve clarity and flow. Since this opens the Results and interpretation section, first, clearly define the two events being discussed. Then address

the apparent contradiction between describing the events as both "understudied" and "well defined." Consider reorganizing the information to: First establish these events as documented in deep-sea records. Then summarize the existing research. Finally, identify this research gap in siliceous microfossils and biosiliceous fluxes and explain how your new dataset will contribute. This structure will better frame your contribution within the presented existing literature.

This paragraph has been modified.

Line 87: It would help this paper to have a defined results section and a defined discussion section. This mixed discussion is at times hard to follow. You measured isotopes in samples 700B, 752A, and 1121B. The text mentions the data set does not cover the LDE. Was this missed? Do you compare your record to the isotopes from Hollis et al., (2024) and there are similarities where there is overlap so you decided to move forward with this combined approach?

Unfortunately, only samples from Hole 700B cover the LDE. This is because we did not have samples from Holes 752A and 1121B for this period. We discovered that Hollis et al. (2014) had already produced stable isotope records covering the LDE at Hole 1121B. We therefore decided to compare our records and use Hollis et al. (2014) data to improve our manuscript.

Line 88 and 89: Is this using the age model? Or do you note the LDE based on something else? Should these studies be cited?

The LDE is defined in the literature as occurring at ~62.2 Ma, so this is what we used as a reference in this study, but we removed this sentence and gave more precision on the definition of the LDE in the paragraph opening the Discussion section, when restructuring the paper.

Line 96 and 97: Might be good to discuss this further. What is your evidence for calibration issue? Is there anything in the core section description that might also highlight some potential issue?

The age-depth model is very coarse for this site and there is nothing in the core description that could explain this issue. We have elaborated a little more on the options in the text to explain why this peak occurs earlier than the others, including a possible link with palaeogeography.

Line 99 and 100: "Hole 700B…with 1051A)." Is this in the supplemental discussion? It seems like this would be better discussed in a results section. Then you can discuss in the broad paleoceanographic section what these differences mean with respect to biogenic silica accumulation.

We have moved this part to the Results section.

Line 105: Which Figus et al.?

Same as for line 36.

Line 106: Which Figus et al.?

Same as for line 36.

Line 106/107: "Biosiliceous content analyzed in present study." Is this something you have analyzed for this work?

We checked the content of the samples pre-silica extraction during the revisions. The results are available in Table 2.

Line 108: What is the resolution compared to the million years compared to your other work? How is it more precise? And "Figus et al. (2024)"?

Resolution is not regular throughout the records, but it ranges from tens to hundreds of thousands of years.

Line 113: change "climaic" to climatic

This has been corrected.

Opening sentence of Line 114/115: This sentence could use some restructuring. How was this identified in this study cited? How was it identified better with precise characteristics?

This sentence has been restructured.

Line 115/116: Citation for hyperthermal event and the originally described carbonate dissolution event. Is this from the report? A new study? Petrizzo (2006)?

This sentence has been deleted to improve the flow of the text and avoid redundancy with the introduction.

Line 119: change "foraminera" to "foraminifera"

This has been corrected.

For the 3.2.1 A climatic and/or biotic event?: You end this paragraph discussing Hollis et al., (2014) and the potential for 1121B to represent cooling linked with enhanced upwelling and marine productivity. Are there papers to be cited for this? Also, how do your biogenic silica records either confirm or deny or add to the complexity of characterizing the ELPE as a climatic or biotic event?

We have not found any other authors who discuss or agree with Hollis' theory. What our paper shows is that in terms of biogenic silica, if the ELPE has an impact on Si, it probably did not take place at the same time everywhere, or several small events may have occurred.

Line 134: "but onshore sites disagree"; however, onshore sites appear to disagree" or something to connect these ideas.

As we have used 'although', there seems to be no need to use 'however'.

Line 134 to 136: I note that you call all these Sites deep-sea, but given you end the paper by talking about specific oceans I would consider in addition to ages providing information on the ocean basin. "~58.4 Ma in the Northwest Pacific" or "58.9 Ma South Atlantic Sites"

This has been corrected.

Line 138 and 139: Might be good to include these onshore sections on the location figure 1.

We have added these onshore sections as well as the deep-sea sites used to constrain the ELPE in other studies.

Line 150 and 151: Is this data not available for plotting for comparison with the rest of the records?

The difference in resolution between the published data and ours was too significant to produce a figure that would have been clear for comparison.

Line 164/165: Might be better to have a discussion figure by ocean basin with isotopic shifts and biosiliceous fluxes. Right now, jumping between the two figures doesn't clearly highlight this. Or if you wish, maybe a transparent box in these intervals to draw our attention to the shift in this time range to aid in this comparison.

We have modified Figs 2 and 3 to have one figure with bioSiO2 flux and stable isotope data per ocean basin (Fig. 2). Fig. 3 now represents the %CaCO3.

Line 166 to 172: What do the biogenic silica values you calculated/the fluxes indicate about this time step? Is inflow of cold-water masses corrosive to carbonates why there are peaks in biogenic silica? Is there some other explanation?

Cold waters are generally nutrient-rich and if sediments are not calcareous, they are often siliceous.

Line 185 to 190: All findings/analyses should be discussed here. The oxygen isotopes from sites... tell you the carbon isotopes from sites aid in interpreting…. The biogenic silica values and fluxes. What is uncertain stratigraphy? Can it be improved?

We have deleted the part on uncertain stratigraphy and reworked the conclusion.

[Figure]

**Responses to Anonymous Referee #2:**

Review of Figus et al.

This manuscript reports on two poorly studied hyperthermal events (LDE and ELPE) within the Palaeocene. Whilst most studies have focused on calcareous microfossil records, the present investigation deals with biosiliceous fluxes, supported by geochemical data. The Palaeocene is a notoriously bad time interval for diatom preservation, so any new information that can be revealed will be of interest to both palaeoceanographers and micropalaeontologists.

The manuscript is well-written and illustrated, however, some minor to moderate revision is recommended, after the following points are addressed/considered.

Line comments:

Line 2: outside of the prominent …

This has been corrected.

Line 15: CO2 -> $CO_2$ - use subscript for 2

This has been corrected.

Line 36 and elsewhere: Figus et al. - is this a submitted or accepted (but not yet published) manuscript?

We sincerely apologise for the confusion with 'Figus et al.', it seems that LateX did not take into account the full reference during the final formatting of the manuscript. Figus et al. in the text refers to the preprint 'Controls on Palaeogene deep-sea diatom-bearing sediment deposition and comparison with shallow marine environments' by Figus et al. (2024). This has been corrected.

Lines 36-37: Figus et al. (unpublished?) report that, there is no apparent biosiliceous flux response to Palaeocene hyperthermals - which is not surprising, given the fact that siliceous plankton are normally associated with, for instance, cooler waters and upwelling. Furthermore, they state that shallow marine diatomites in epicontinental seas suggest a link with some hyperthermals. However, I presume the authors did not observe samples of epicontinental seas in this new study or none were available of the relevant Palaeocene age?

Unfortunately, the confusion with 'Figus et al.' has made it unclear. I previously published the results of a global compilation of Palaeogene shallow marine diatomites (Figus et al., 2024a), and another paper, on a global compilation of Palaeogene deep-sea diatom-bearing sediments, is currently in typesetting with Biogeosciences (Figus et al., 2024b). What the shallow marine diatomite paper shows is that there is an abrupt increase in the number of diatomites formed in epicontinental seas during the PETM, but the results are different in the deep-sea paper, in which the number of diatom-bearing sediments does not respond to climatic events. This is why we

decided to take a closer look at biosiliceous fluxes in the deep-sea in the Palaeocene. We wanted to see exactly what happened with biosiliceous sediments during climatic/biotic events of the Palaeocene and we focused on the ELPE and LDE because no biosiliceous flux data have ever been published for these events.

Lines 49 and 51: it says, sites contain abundant and/or well-preserved diatoms as documented in DSDP and ... (ODP) reports, and a few lines later, also contain abundant radiolarians over the entire section analyzed - were these samples checked by the authors? If so, were quantitative measurements made of siliceous microfossils/g to compare with the biogenic opal determinations?

No quantitative measurements were made, but during the revisions of the manuscript, we checked the microfossil content of 10 selected samples around the LDE and ELPE. The results for the LDE are available in Table 2 and the full results are available in the Supplementary Material Table S2.

Table 1: It would be helpful if the table included the water depth (seabed depth). Given that the samples are from DSDP and ODP holes and sites, one would presume that they were drilled in relatively deep waters. If so, this would have an effect on the abundance (biosiliceous flux) of each microfossil group and composition of the siliceous microfossil assemblages (i.e., with perhaps a higher ratio/abundance of diatoms in shelf/upper slope sediments, than in lower slope sediments).

We have added the water depths to Table 1.

Lines 56, 88, 131, 145, Figs 2 and 3 captions: Holes not Sites - you need to be consistent with the use of Hole and Site throughout the manuscript.

This has been corrected.

Line 81: I think the Results and interpretations should be separated to make it clear what is being presented as new and what is being compared with the literature.

We have divided the Results and Interpretation section into two sections.

Line 95: site -> Site

This has been corrected.

Line 110: In the new study, the authors report on a biosiliceous flux peak in the LDE, and suggest that it may be caused by radiolarians not diatoms - what is this assumption based on? The authors wrote 'suggest' - so does this imply that the authors do not know which siliceous microfossil group dominates in each sample? If the composition is known, maybe this data could be shown. If the peak is caused by radiolarians rather than diatoms, what does that mean, ecologically or oceanographically?

This assumption was based on the comparison of the results of this study with those of Figus et al. (2024b), and is now confirmed by our microscopic observation of samples corresponding to the

LDE biosiliceous peak during revisions of the manuscript. This means that, unlike diatoms, radiolarians seem to respond to Palaeocene climatic events.

Line 113: climaic -> climatic

This has been corrected.

Line 119: foraminera -> foraminifera

This has been corrected.

Line 163: it says, probably due to the presence of clay minerals derived from ash alteration. Why 'probably'? Has this been checked? Are there any ash layers reported for these sites? Is it possible that, instead, these are authigenic clays from biosilica-rich sediment alteration?

This information was reported by Shipboard Scientific Party (1989). As the presence of clay minerals derived from ash alteration was not reported in the core in which we had a peak in biosiliceous flux potentially corresponding to the ELPE, we did not focus on it.

Line 236: Clima? or Climate?

This has been corrected.

Line 240: approch -> approach?

This has been corrected.

Fig.1: site/hole locations

This has been corrected.

Supplement review
Fig. S1 captions, Fig. S2: sites -> holes    Site -> Hole
Tables S1 and S2: Site/Hole

This has been corrected.